# Biochar Production and Characteristics, Its Impacts on Soil Health, Crop Production, and Yield Enhancement: A Review

**DOI:** 10.3390/plants13020166

**Published:** 2024-01-08

**Authors:** Shahbaz Khan, Sohail Irshad, Kashf Mehmood, Zuhair Hasnain, Muhammad Nawaz, Afroz Rais, Safia Gul, Muhammad Ashfaq Wahid, Abeer Hashem, Elsayed Fathi Abd_Allah, Danish Ibrar

**Affiliations:** 1Colorado Water Center, Colorado State University, Fort Collins, CO 80523, USA; 2Department of Agronomy, MNS-University of Agriculture, Multan 64200, Pakistan; 3Department of Biological Sciences, Superior University, Lahore 54000, Pakistan; 4Department of Agronomy, Arid Agriculture University, Rawalpindi 46300, Pakistan; 5Department of Agricultural Engineering, Khwaja Fareed University of Engineering and Information Technology, Rahim Yar Khan 64200, Pakistan; 6Department of Botany, Sardar Bahadur Khan Women’s University, Quetta 87300, Pakistan; 7Department of Agronomy, University of Agriculture, Faisalabad 38040, Pakistan; 8Botany and Microbiology Department, College of Science, King Saud University, Riyadh 11452, Saudi Arabia; 9Plant Protection Department, College of Food and Agricultural Sciences, King Saud University, Riyadh 11452, Saudi Arabia; 10Crop Science Institute, National Agricultural Research Centre, Islamabad 45500, Pakistan

**Keywords:** biochar, crop productivity, organic amendments, microbial activity, soil fertility, sustainable agriculture

## Abstract

Rapid urban expansion and a booming population are placing immense pressure on our agricultural systems, leading to detrimental impacts on soil fertility and overall health. Due to the extensive use of agrochemicals in agriculture, the necessity to meet the expanding demand for food has also resulted in unsustainable farming practices. Around the world, biochar, a multipurpose carbonaceous material, is being used to concurrently solve issues with enhancing soil fertility, plant growth, and development under both normal and stressful circumstances. It improves water retention, fosters nutrient absorption, and promotes microbial activity, creating a fertile environment that supports sustainable and resilient agriculture. Additionally, biochar acts as a carbon sink, contributing to long-term carbon sequestration and mitigating climate change impacts. The major benefit of biochar is that it helps the adsorption process with its highly porous structures and different functional groups. Understanding the elements involved in biochar formation that determine its characteristics and adsorptive capacity is necessary to assure the viability of biochar in terms of plant productivity and soil health, particularly biological activity in soil. This paper focuses on the development, composition, and effects of biochar on soil fertility and health, and crop productivity.

## 1. Introduction

Promising strategies for ensuring food security for a growing world population include increasing production per unit area and sustainably improving agricultural productivity. Research is being conducted around the world on the potential benefits of using biochar as a solution of food security by enhancing crop productivity [1,2,3]. The application of biochar is coherent with modern green development concepts as it plays critical roles in maintaining ecosystem balance, controlling soil pollution, and the sustainable development of the agricultural environment [4]. Biochar is a carbon-rich and porous material produced through the thermal decomposition of biomass, such as plant residues, agricultural waste, or wood, under controlled oxygen-limited conditions. This process, known as pyrolysis, involves heating the biomass to high temperatures in the absence or near absence of oxygen, preventing complete combustion. The result is a stable form of charcoal that contains a well-structured network of pores called biochar [5]. It has attracted a lot of attention in the last decade due to its numerous benefits in various fields, including soil health, agriculture, wastewater treatment, and climate change [6]. Improving soil quality, productivity, and carbon sequestration while reducing greenhouse gas emissions (CO_2_, N_2_O, CH_4_) are clear impacts of using biochar [7,8,9]. According to current knowledge, plant organisms can help soil absorb about 12% of all anthropogenic carbon dioxide emissions from land use change (0.21 Pg C) every year. According to numerous studies, the use of biochar improved soil microbial activity and physical properties such as pH, cation exchange capacity (CEC), pore size distribution, bulk density, soil structure, soil organic carbon (SOC), and soil water holding capacity [10,11,12]. In addition, biochar can prevent nutrient losses through leaching, increase the bioavailability of soil nutrients, and bind hazardous substances in contaminated soils [13,14].

The sustainability of biochar application to soil remains uncertain. Research indicates a wide-ranging temporal spectrum, with effectiveness spanning from a few years to potentially millennia [15]. The production process (including feedstock type and pyrolysis temperatures), weather conditions, soil structure, and other environmental factors affect the stability of biochar in soil [16]. Depending on location, deployment is not expected to have the same impact on soil fertility and carbon stability. Numerous studies, even laboratory studies, have shown the beneficial effect of using biochar and organic amendments on yields [17,18,19,20]. Several studies conducted in temperate regions showed no negative effects of biochar on the yield of various crops [21,22,23,24]. Furthermore, most currently available information on biochar production, its properties and its effects on soil is based on small laboratory experiments (in pots or plots) and lacks field studies. Another problem of biochar research is synchronizing the results of experiments on different geographic heterogeneities such as different soil types, climate, and land use. Therefore, it is difficult to effectively compare different approaches. Furthermore, tropical regions are still underestimated in the research, despite severe pressures on the natural resources of these regions.

Agriculture and the development of climate protection plans are currently the two most important large-scale uses of biochar [25]. This review summarizes knowledge of soil properties (physical, chemical, and biological), crop production, and the role of biochar in climate change from field trials over the past decade. In addition, an attempt is made to describe the differences in the effects of biochar between tropical and temperate regions. It also emphasizes the need for extensive field research on biochar to demonstrate impact and the importance of its regional impacts.

## 2. Biochar Production

### 2.1. Thermal Conversion

With the growing interest in using biochar for multiple purposes, the conversion of biomass to biochar is also growing. Biochar is usually produced by thermochemical conversion. Thermochemical conversion processes include roasting, gasification, hydrothermal carbonization, and pyrolysis [26,27]. To obtain the highest possible yield of biochar, the production method must be adapted to the type of biomass and process parameters such as heating rate, temperature, and residence time. These conditions are critical because they affect the physical and chemical properties of biochar during the manufacturing process. Since the process results in biomass loss, the form of biochar produced from plant biomass depends on process conditions.

### 2.2. Gasification

A thermochemical process called gasification converts carbon compounds into gaseous byproducts. In the presence of carbon monoxide (CO), carbon dioxide (CO_2_), methane (CH_4_), hydrogen (H_2_), and gas-generating agents such as oxygen, air, and steam, synthesis gas H_2_ with a high temperature and containing trace hydrocarbons is produced. It should be noted that the reaction temperature is the main factor affecting the synthesis gas production. Studies have found that, as temperatures increase, more CO and H_2_ are produced, while less CH_4_, CO_2_, and other species such as hydrocarbons are produced [28].

### 2.3. Torrefaction and Flash Carbonization

A new biochar production method is roasting. Due to the low heating rate, it is called mild pyrolysis. In an inert anaerobic environment at 300 °C, multiple degradation mechanisms remove the oxygen (O_2_), moisture, and CO_2_ present in biomass [29]. The roasting process changes the properties of the biomass, including particle size, moisture content, surface area, heating rate, and energy density. The combustion process includes evaporation; biomass is evaporated at a maximum temperature of 260 °C and is wet-baked for about 10 min; hot water carbonization is the process of heating water to 180–260 °C. Next is oxidative roasting. In this process, biomass is treated with an oxidizing agent (such as flue gas) to generate heat energy. This thermal energy is used to generate the desired temperature. The mechanism of the roasting process is an incomplete pyrolysis process, and the reaction parameters are a temperature of 200–300 °C, a holding time of less than 30 min, and a heating rate of less than 50 °C/min, air. Heating, drying, toasting, and cooling are some of the steps in the dry roasting process. Also in this case, the pre-drying and post-drying processes can be separated.

### 2.4. Hydrothermal Carbonization

Hot water carbonation can produce biochar at temperatures between 180 and 250℃ and is a cost-effective method [30]. To differentiate them from products generated through dry methods like pyrolysis and gasification, those produced through hydrothermal processes are termed hydrocarbons [31]. The closed reactor is filled with a mixture of water and biomass. The hydrolyzate must go through a series of processes such as dehydration, cracking, and isomerization to produce 5-hydroxymethylfurfural and its derivatives. Furthermore, the reaction process leads to the formation of hydrocarbons through intramolecular condensation, polymerization, and dehydration [32]. The high molecular weight and complex nature of lignin complicates this mechanism. The dealkylation and hydrolysis processes leading to phenolic compounds such as phenols, catechins, and eugenols are the first steps in lignin destruction [33]. The final product carbon is formed by repolymerization and the crosslinking of intermediates. The lignin components that are not dissolved in the liquid phase are transferred into the hydrochar-like pyrolysis process.

## 3. Biochar Characteristics

Biochar, a multifaceted soil amendment, undergoes thorough characterization to assess its potential for pollutant removal and its diverse applications. Environmental impact prediction becomes an intricate process through structural and elemental analyses. Key revelations include the pH-dependent property of biochar functionality with metal interactions and its role as an efficient soil pollutant purifier. The pH of biochar is influenced by several factors, and understanding these factors is crucial for its effective application in various contexts. One significant determinant is the feedstock material used for biochar production. Different biomass sources have distinct chemical compositions, leading to variations in the pH of the resulting biochar. For instance, feedstocks with high ash content may produce biochar with a higher pH, as ash components can be alkaline in nature [34]. Additionally, the pyrolysis temperature during biochar production plays a vital role in pH control. Higher pyrolysis temperatures often result in biochar with increased alkalinity due to the oxidation of acidic functional groups [35].

Brewer et al. [36] unravel the secrets of biochar through elemental analysis and surface structural wizardry. Assessing biochar stability involves evaluating its resistance to biotic and abiotic soil degradation, often predicted by pyrolysis temperature. However, this method’s imprecision prompts the exploration of alternatives. Techniques include the quantitative identification of carbon structures, stability quantification via thermal and chemical processes, and the incubation and modeling of biochar in soil. The unique presence of carbon in biochar, specifically its crystalline and amorphous phases, provides insights into stability. Incubation and modeling, despite their effectiveness, are costly and time-consuming, pushing researchers to explore innovative approaches like 14-C radioisotopes for improved climate change mitigation.

Basic functional groups like carboxylic acid and hydroxyl impact biochar adsorption capacity. Surface functional groups vary with temperature and biomass. Fourier transform infrared spectroscopy (FTIR) and nuclear magnetic resonance (NMR) techniques play crucial roles in studying these groups. The high surface area and porosity of biochar, formed during pyrolysis, make it highly absorbent. Pore size classification and scanning electron microscopy (SEM) aid in understanding structural properties. Activation processes enhance surface area and porosity, which are critical for biochar efficacy. These diverse characteristics, analysis techniques, and insights contribute to a comprehensive understanding of biochar potential applications and its effectiveness in environmental solutions.

## 4. Biochar Impacts

The application of biochar is one of the sustainable approaches to improving the physical and chemical properties of soil, and the quality of produce and crops yield [37]. Furthermore, biochar has proven to be efficient in different applications, particularly soil amendment for crop production and the removal of pollutants from the contaminated water and soil environments [38]. The effects of biochar on soil health, physical and chemical properties, nutritional status, biological activities of the rhizosphere, and crop growth, development, and yield are examined in the section that follows.

### 4.1. Soil Health

In terms of agriculture, soil health refers to the soil’s capacity to sustain and promote plant development and production [39]. If a soil can supply the water and nutrients needed to support plant growth while being free of hazardous substances that could hinder it, it is considered fertile soil [40]. The physical, chemical, and biological characteristics of the soil determine its fertility [13]. Low soil fertility is a widespread issue in several regions of the world [41]. For instance, soil in arid and semi-arid regions typically has a low capacity for storing water for most crops and an insufficient supply of nutrients [42]. Due to excessive rainfall, limited cation binding capacity, and rapid loss of vital nutrients from the topsoil, rainforest regions struggle to maintain agricultural output. Additionally, the frequent breakdown and relatively high temperature lead to significant rates of soil organic matter (SOM) mineralization [43]. Therefore, the success of a soil management system depends on correct SOM levels and nutrient life cycles. The use of biochar has the potential to greatly enhance soil health. Whether the application of biochar has the potential to enhance the productivity of degraded pastureland has also been investigated [44]. According to Brtnicky et al. [45] and Joseph et al. [46], biochar has a wide range of possible uses that are likely to: (i) improve soil properties; (ii) increase soil protection and water retention; (iii) prevent soil degradation and losses; (iv) increase nutrient content and sequestration in soil; (v) attenuate the impact of potentially toxic substances; (vi) promote the well-being of organisms in the soil environment; (vii) enhance plant growth and biomass production and quality; and (vii) increase the crop yields and profits of the agricultural sector.

### 4.2. Physical Properties

#### 4.2.1. Bulk Density and Soil Compaction

Soil weight and density have a direct and significant effect on soil properties. Another important factor affecting root development is the physical stress of soil compaction, which usually occurs at a depth of about 30 cm in the soil. When the bulk density exceeds 1.7 g cm^−3^, insufficient aeration and stamina hinder root growth [47]. The reduced soil pore diameter inhibits root growth, as roots cannot grow through soil pores smaller than the root cap diameter [48]. Low porosity and low oxygen diffusion in compacted soils can lead to anoxic conditions and slow plant growth. When planted in hard soil, soybean (45%) and maize (14%) yields decreased compared to normal soil [49]. The use of biochar can mitigate these effects and improve the physical properties of nutrient-poor and degraded soils. The use of biochar has been shown to reduce soil compaction by more than 10% [50]. Furthermore, previous studies have shown that adding biochar to dry soils reduces soil density when the soil is physically loose [8,51]. According to a recent study, adding 4% biochar to locally eroding tropical soils (sandy clay) for a year increased soil density and decreased bulk density by 5%. Additionally, using 1% compost with 4% biochar resulted in a 16% reduction in bulk density and an 8% increase in porosity. Increased use of biochar (6%) and compost (1%) reduced bulk density and further increased porosity to 16% and 22%, respectively [52]. A review of studies on 22 different soils showed that biochar addition reduced soil bulk density by 3–11% (average 12%) and increased porosity by 1–64% [53]. Increasing soil porosity and reducing bulk density improves water, air, and heat transport in soil [54,55]. As mentioned above, the degree of variation in physical properties depends on the amount of biochar used and the type of soil.

#### 4.2.2. Porosity and Water Holding Capacity

Changes in soil porosity were mainly due to internal pore structure (intrapores), biochar morphology and application rate, pores between biochar and soil particles (interpores), and soil changes due to adsorption and particle size distribution [56,57]. These factors determine the increase or decrease in porosity, water retention, hydraulic conductivity, and respiration in biochar-rich soils. Numerous field studies have demonstrated that the use of biochar can improve soil porosity and water retention [58,59]. According to a meta-analysis of 74 articles [11], biochar benefits from an increased soil porosity, water retention, and saturated hydraulic conductivity of 8.4, 15.1, and 25, respectively. The internal porosity of biochar (biochar pore size <10 µm) determines its ability to store water in the soil. This is because biochar replenishes water as it is removed by gravity, trapping water in pores, reducing hydraulic conductivity, and increasing water retention [60]. Wong et al. [61] showed that the degree of compaction affects the degree of aeration of clays in response to biochar addition. The addition of biochar to soil helps maintain its structure and compaction [62,63]. It has been found that soil water availability decreases with increasing biochar additions, possibly due to the hydrophobic nature of biochar [64]. Generally, biochar synthesized at temperatures below 450 °C exhibits hydrophobicity [65]. Furthermore, it has been suggested that biochar can act as a binder to strengthen soil aggregates, increase water absorption, and increase porosity [52]. Due to the use of 6% biochar in clay soil, biochar promoted the development of macroaggregates, changed the pore structure of the soil, and stabilized the soil [66]. However, when biochar is applied, large aggregates smaller than 0.25 mm block the pores on the surface of the biochar (inner pores), thereby reducing the size of the mesopores (pores between the biochar particles and the soil). Surface sealing reduces the porosity of the soil and reduces moisture. The increased leaching of dissolved organic carbon is the result of abiotic surface reactions of biochar that alter the surface chemistry, particle size distribution, hydrophobicity, and physical solubility under the action of water [65,67]. To change the physical properties of soil treated with biochar, biochar breaks down over time, plugging pores and forming aggregates. Manure, grass, corn stover, and other cellulosic foods do not break down into biochar as easily as lignin-rich wood. In addition, biochar produced at temperatures below 500℃ is prone to structural degradation. The addition of biochar resulted in unstable changes in soil porosity and particle size distribution. These changes mainly occurred in sandy soils [68].

### 4.3. Soil Chemical Properties

The use of biochar has shown significant potential to enhance the chemical properties of soil. The application of biochar and other organic amendments has been shown to improve agricultural productivity by increasing cation exchange capacity and nitrogen content in soil [69,70].

#### 4.3.1. Soil pH

Due to its high alkalinity, strong buffering properties, and the presence of functional groups, biochar can be used to combat soil acidification [51,71,72]. In addition, it increases soil pH, increases plant nutrient availability, and releases cations such as potassium, magnesium, calcium, and sodium from charcoal [54,73,74]. After 4 years of biochar application (20 tons ha^−1^), soil pH increased from 3.89 to 4.05, confirming its long-term beneficial effects [71]. In Sumatra, Indonesia, another study using 20 tons of ha-1 biochar found an increase in soil pH from 3.9 to 5.1 and a decrease in soil Al^3+^ concentrations (toxic to plant growth) [75]. The use of biochar in banana cultivation was studied and showed improvements in soil pH and potassium uptake, but no significant effect on fruit yield [76,77]. In clay, there is a slight increase in pH due to the high initial CEC, which translates into a higher buffering capacity. In a three-year field study, biochar did not change soil-dissolved organic nitrogen, dissolved organic carbon, or NH^4+^ or NO_3_ levels, but eventually completely neutralized their alkalinity [78].

#### 4.3.2. Salinity and Sodicity

Some soils contain significant endemic salinity due to saline irrigation and the use of chemical fertilizers. Turf characteristics include higher EC, ESP, and pH. These soils reduce agglomeration through mechanisms such as hardness, clay swelling, and dispersion. Furthermore, high soil salinity can lead to osmotic stress and drought, which slow down the microbial and biochemical activities in the soil [79]. Saline soils contain less organic matter and are therefore less structurally stable. Therefore, adding organic additives such as biochar to soils can reduce salt stress and enhance plant growth in these soils [80]. Salt and fertility problems in these soils can be reduced by adding acidic (low pH) charcoal [81]. Biochar mitigates sodicity and salinity in soils through multiple mechanisms. Firstly, it enhances soil structure, promoting better water infiltration and reducing the impact of high sodium levels. Secondly, biochar’s cation exchange capacity helps balance soil ions, mitigating salinity stress on plants. Additionally, it facilitates microbial activity, fostering the growth of salt-tolerant organisms that contribute to improved soil health and salinity reduction.

#### 4.3.3. Cation Exchange Capacity

The ability of soil to retain cations in an exchangeable form that is available to plants is called cation exchange capacity (CEC), and it increases in proportion to the amount of surface negative mineral charge and soil organic matter (SOM) [64]. Soils with a high CEC can retain plant nutrients and cations on the surface of biochar, humus, and clay, allowing nutrients to be retained rather than leached or absorbed by plants [54,58]. Due to the high buffering capacity of the soil due to its high CEC content, the addition of alkaline or acidic chemicals had little or no effect on soil pH [77]. Freshly added biochar undergoes surface oxidation reactions on contact with soil water and oxygen, increasing the CEC and net negative charge. The presence of many reactive functional groups (COOH, OH, CO, C-O, N, siloxane), some of which are pH-dependent, contributes to the high reactivity of the biochar surface [82]. A two-year field study in the Indonesian highlands of East Java (tropical region) showed an increase in CEC following biochar application. This is due to the oxidation of carboxyl groups and the high negative surface charge of phenolic biochar [83]. However, in a field study, green biochar residues (10 t ha^−1^) had no effect on CEC and iron salts (in highly modified acidic soils) [84]. Mild non-calcareous soils showed a strong increase in CEC after bioburden, although calcareous soils also showed this trend [75,85,86].

### 4.4. Nutrient Offering and Retention

The monsoon season, particularly in tropical regions, leads to nutrient loss (food waste brought in from the outside), accelerated acidification of agricultural soils, lower crop production, and increased fertilizer demand. A practical way to prevent nutrient loss is to add biochar to soil [87]. Some long-term benefits of adding biochar are the slow release of nutrients from additional organic matter, better stabilization of organic matter, better utilization of nutrients, and the retention of cations and CEC [88]. Considering soil alone, Brazilian pepper biochar significantly reduced total phosphate, nitrate, and ammonium levels in wastewater by 20.6%, 34%, and 34.7%, respectively. Like the 34% and 14% reduction in nitrate and ammonium leaching, respectively, peanut shell biochar showed the same results [89]. Several field studies have shown that the addition of biochar to agricultural soils significantly reduces the soil’s ability to leach N, NO_3_, K, P, Mg, Na, and Ca [90,91,92]. Another field trial in Zambia increased the pH of tropical soils by adding corn stover biochar, which also increased the amount of P available to plants and contributed K+ directly to the soil [93]. In a banana plantation in Tamil Nadu, India, a two-year field trial tested the ability of rice husk biochar (10 t ha^−1^), as an additive, to improve the fertility and moisture content of soil in India. Through this adjustment, the contents of P, K, Na, Mg, C, and N in the soil have been greatly improved. For tropical soils with high mineral content, the authors recommend using biochar above 10 t ha^−1^ [93]. In addition, biochar increases soil nitrogen content and reduces nitrogen leaching [51,94]. Several field studies have reported nitrate retention and ammonium adsorption following the use of kale [54,95]. Field trials of tomato transplants (14 t ha^−1^) using wheat bran biochar significantly increased the availability of SOC, P, K, Mg, and NH^4+^ by reducing the dependence on external water and fertilizers [96]. According to Liu et al. [97], the application of biochar could not only improve carbon utilization efficiency by the soil microbial community, but also the soil organic carbon sequestration potential in paddy soil can be enhanced by the presence of biochar in soil long term.

### 4.5. Soil Biological Properties

Diverse microbial populations live in soil and are constantly changing according to soil properties, climate, and land management techniques [98]. Soil microbial diversity and activity are closely related to organic carbon concentration, soil nutrient cycling, and plant productivity [99,100]. The effect of biochar addition depends on soil type, biochar quality and dosage, and on the activity of microorganisms in the soil [101,102]. A meta-analysis showed that biochar modification increased denitrification genes (nirS, nirK, nosZ) and ammonia oxidizing archaea by 25.3%, 32.0%, 14%, 6% and 17.0%, respectively [103]. Biochar supports the activity of soil bacteria by providing carbon and nutrients for their growth. In addition, it provides a favorable growth environment [94,104]. In addition, biochar increased the soil’s ability to absorb acid changes and reduced pH changes in microhabitats found on charcoal pellets [101].

Biochar (BC) was used at doses of 0 (BC0), 10 (BC10), 20 (BC20), and 30 g/kg (BC30). Microbial biomass carbon (MBC) and microbial biomass nitrogen (MBN) were assessed [105]. They reported that MBC did not change significantly during the growing season. The variability of MBC was relatively large under different biochar addition rates. Biochar significantly increased MBC by 15.2–71.8% during vegetation in a 0–10 cm thick layer compared to BC0. Biochar also significantly increased MBC in the 10–20 and 20–30 cm layers, and MBC increased with biochar. Unlike MBC, MBN exhibits high variability across growing seasons. Biochar had a significant effect on MBN; the growth period and soil depth also contributed to differences in MBN. BC10 significantly increased MBN at stage V6, but BC20 and BC30 had no significant effect on MBN compared with BC0 in the 0–10 cm layer at stage V6. Biochar application significantly increased MBN over the remaining growing season in the same soil layer. In formations 10–20 cm thick, BC10 and BC30 significantly reduced MBN by 14.6% and 13.6%, respectively. Significant decreases in MBN were also observed in BC20, while significant increases were observed in BC10, BC20, and BC30 compared to BC0. Biochar generally reduced MBN in 20–30 cm thick soil layers, regardless of the growing season.

They also showed that biochar dosage and growth period had a significant effect on the ratio of MBC to MBN, which ranged from 1.7 to 9.4. Typically, MBC/MBN in the soil layer increased from 0 to 10 cm due to the addition of biochar. While ratios in other steps vary depending on the amount of biochar used, biochar increases MBC/MBN in the R1 step compared to BCO in the 10–20 cm layer. In contrast, biochar significantly increased MBC/MBN in the 20–30 cm layer throughout the growing season. Microbial ratios varied between 2.5% and 4.0%, remaining largely constant across different soil layers and growth cycles. Biochar had no significant effect on microbial proportions, although BC10 and BC30 were significantly lower compared to BC0. According to Filho et al. [106], the type, amount, and interaction of biochar has a significant impact on microbial biomass carbon. Since coffee biochar contains unstable organic carbon and has pores that can store large amounts of air and oxygen more quickly, which are crucial for the utilization and growth of microorganisms, its application to sandy soils increases the rate of microbial biomass growth and carbon content [107]. However, due to its high aromaticity and low polarity, biochar can resist microbial degradation [108]. The large surface area of biochar provides an ideal microhabitat for microbial communities and is another reason for the increase in soil microbial biomass [107]. Tan et al. [109] found that using biochar together with organic fertilizers significantly increased soil microbial biomass carbon (MBC) compared to using organic fertilizers alone.

It has been speculated that biochar may influence soil microbial populations, which may also affect soil fauna due to cascading effects. The physical fragmentation of organic waste, including biochar, and changes in soil structure, as well as the direct impact of biochar on fauna, affect microbial communities. If biochar creates a favorable environment for soil microbes, then wildlife may be attracted to these microbe-rich environments. However, there are few experimental studies to determine the response of soil fauna to the addition of biochar to soil and the main factors influencing their behavior. Therefore, the behavior of two mesofauna in response to the addition of saline biochar (slow pyrolysis at 600 °C) to temperate clay was assessed as associated with changes in microbial activity and biomass after 17, 31, and 61 days of pre-incubation. With the increase of biochar concentration at different pre-incubation times, the microbial biomass increased by 5–56%, and the activity increased by 6–156%. Using different amounts of biochar, microbial biomass was unchanged or increased by up to 15% during the incubation period, but microbial activity continued to decrease (by 70% to 80% at day 61). In general, when a different, unchanged substrate was used instead of biochar, springtails did not show avoidance or preference for it, but frequently exhibited avoidance behavior [110].

Soil enzymes that can revolutionize soil microbial ecology and beneficial bacteria that are critical to agriculture have all been improved with biochar. Soil ecology is closely related to the number of microorganisms in the soil [111]. Biochar can increase microbial diversity by providing substrates and habitats for soil microbial communities. In addition, soil microbes are sensitive to changes in the soil environment. By creating microenvironments, providing unstable organic molecules for bacterial growth, and activating nutrients, soil particles can interact with biochar to influence microbial populations [112]. During biochar application, the particles and pores create a habitat that can serve as a means of infiltration for fungi and numerous filamentous microorganisms [113]. Some biochars are rich in sugars and yeasts that promote the growth of Gram-negative bacteria (*Pseudomonas putida*) and soil fungi (*Pythium* sp.). Under field conditions, the use of biochar in brown soils resulted in stronger and more vigorous and aggressive bacterial populations than fungal populations. Furthermore, Saxena et al. [114] found that, in four different soils, the fungi-to-bacteria ratio was significantly lower in soils rapidly treated with pyrolytic biochar after an incubation period of 365 days.

Compared with pyrolysis temperature and feedstock type, organic carbon in biochar has a significant effect on many soil biological properties, including nitrogen mineralization and immobilization, carbon in microbial biomass, enzyme activity, and soil biocarbon dehydrogenation enzyme activity. Gram-negative bacteria benefited less from biochar compared to Gram-positive bacteria, suggesting that biochar is often deficient in organic compounds that break down into the small molecules necessary for the growth and reproduction of Gram-negative bacteria, such as proteins, amino acids, and carbohydrates [115]. While some charcoal-treated soils completely lost their rich microbes, other soils showed increases in the number of monads and actinomycetes. In fact, the pores in biochar are suitable habitats for soil bacteria, protecting them from desiccation and predators, as well as being a source of carbon, minerals, and energy. However, it is questionable whether the use of biochar affects the amount of microbial biomass in different soils. Decreases and increases in carbon content in microbial biomass were found in biochar-treated soils [68]. However, the response may vary and depends on the type of biochar and soil. The addition of 30.0 t/ha of biochar can increase the bacteria count from 366.1 g C g^−1^ to 730.5. As a result, microbial populations increased from 5% to 56% with increasing biochar (0–14%) in corn ovens with different pre-incubation times ranging from 2 days to 61 days. Straw biochar had the highest survival rate, indicating that the biomass type of biochar or biochar material was most important for the function and survival of PSB communities (phosphate-dissolving bacteria).

Biochar has both positive and negative effects, including toxic effects on soil bacteria and highly beneficial effects on interactions with plant roots, biological aging systems, degradation of microbial pollutants, and carbon stabilization through micro aggregation. Its patches can survive bacterial communities and dissolve inorganic phosphate at a rate of 6.86–24.24%. The addition of biochar increased the total number of nitrogen-fixing bacteria, mold, and yeast compared to those without biochar (control). This is because biochar has a positive effect on increasing the number of microorganisms in the soil by providing a useful substrate [116]. Adding biochar to soil can have an impact on soil microbial populations, as demonstrated in the biochar-rich black soils of the Amazon. In contrast, biochar poses a direct threat to soil flora and fauna [117]. Depending on the type of soil and biochar used, biochar can negatively impact soil microbial populations. Polyphenols and phenolic substances may be present in biochar as by-products of organic pyrolysis and may be toxic to soil microorganisms. The use of biochar reduced mycorrhizal and total microbial biomass. The addition of peanut shell reduced mycelial spore length and root colonization of arbuscular mycorrhizal fungi by 95% and 74%, respectively.

The use of mango wood biochar increased phosphorus availability by 208% and 163%, respectively, but decreased the number of mycorrhizal fungi in the soil by 77% and 43%, respectively [118]. On the other hand, the presence of activated biochar was found to increase root colonization, Arabidopsis plant dieback, and *Pseudomonas syringae* symptoms, as the roots desorb antimicrobial chemicals released by the roots to the surface of the activated biochar. In a 3.5-year field study in Tasmania, Australia, the addition of biochar (47 t ha^−1^) resulted in increased microbial populations [119]. Another two-year field study in Australia found that biochar-supplemented soils resulted in an increase in phosphorus-activated mycorrhizas due to the indirect effects of biochar on soil physicochemical properties [120]. In South Sumatra, Indonesia, maize showed increased arbuscular mycorrhizal (AMF) colony formation after the application of mango acacia bark [75]. The study also found that adding biochar had neutral and negative effects on soil microbial activity. In wheat field trials, the addition of biochar (3 or 6 kg/m^2^) did not change the microbial biomass in soil after three to fourteen months [121]. However, a field study applying biochar to the wood of Colombian mango trees (*Mangifera indica*) showed reductions of 43% and 77% of microbial biomass at 23.2 and 116.1 t C ha^−1^, respectively [122]. The production of ethylene or organic pyrolysis by-products, including biochar phenols and polyphenols, can negatively impact soil microbiota, which may be responsible for reduced AMF levels [68,122].

Furthermore, due to its different mechanism of action, biochar can induce different metabolic responses in microbial populations, leading to some taxonomic changes in microbial community composition [123]. Field studies using maize biochar (30 t ha^−1^) at three sites in Europe (West Sussex, UK, Pratoscia, Italy, and Lusignan, France) showed that microbial community composition drastically changes. One year after biochar application, some plants had increased Bacillus and Acidobacteria phyla, while others plants had increased Bacillus and Pseudomonas. In France, however, there was no change in the number of bacteria in this category. Furthermore, biochar treatment influenced fungal diversity in Italy and France, but not in UK samples [124]. Previously, a Chinese laboratory study targeting corn carbon resulted in increases in Proteobacteria, Bacteroidetes, and Actinomycetes, and decreases in Acidobacteria, Chlorobacteria, and Twins. Another field study on sugarcane biochar in Foshan, South China (a subtropical region) showed an increase in bacterial populations and a decrease in actinomycetes and fungal populations [125]. On the other hand, when biochar was added to the lanceolate soil of fir, a significant increase in fungal community diversity was found, but also a significant decrease in bacterial community diversity [126].

### 4.6. Crop Growth, Development, and Yield

Several field studies have explored the impact of biochar on plant growth and the productivity of different crops (Table 1). Its use involves plants with fewer nutrients in degraded soils than in healthy and productive soils [17,18,127,128]. Four-year field trials using rice husk biochar in dry, poor, non-acidic soils in the Philippines and Thailand (tropical climate) resulted in increased seed and water yields, improved K and P availability, and yields of 16–35% [129]. Biochar (2 t ha^−1^) from sorghum and rice husks was used in another field trial in Ghana, where it significantly increased maize yield and improved soil pH, SOC, N, P, and K. Similar results were obtained for durum wheat in the Mediterranean region, with yield increases of up to 30% using 30 and 60 t ha^−1^ biochar [130]. Field studies in South Sumatra, Indonesia, showed that the addition of charred bark improved soil chemistry, created an environment favorable for root growth, and allowed mycorrhizal fungi, maize, and peanut (37 t ha^−1^) Acacia mango to thrive [75]. The application of 10 to 50 t ha^−1^ of eucalyptus biochar to acidic soils in Madagascar (tropical humid environment) improved growth of maize and beans crops, and the yields were significant [131]. In a field study of squash cultivation in Nepal (subtropical environment), it was found that the addition of 0.75 t ha^−1^ biochar and 6.3 m^3^ ha^−1^ cow urine reduced yield by 300% and 85%, respectively [132]. The ability of biochar to absorb and exchange plant nutrients is enhanced by the formation of an organic coating on the outside of the internal pores when saturated with urine. Thus, these studies suggest that biochar supplementation is an effective strategy for increasing yields and is associated with improved nutrient availability and better soil structure and carbon content.

Some field trials produced higher yields in the second year than after biochar application (season 1). A four-year biochar field trial (8 and 20 t ha^−1^) in maize and soybean (alternative) crops in tropical Colombia had increased maize but not soybean yields in the first year [71]. However, in years 2, 3, and 4, there were increases of 20%, 30%, and 140%, respectively, suggesting that biochar has a long-term impact on productivity. Similar results were observed for maize and mustard (crop rotation) in the acid clay soil Raswa, Nepal, with no apparent impact on first-year yields [16]. This may be due to the slow formation of an organic layer on top of the biochar after aging in the compost matrix, improving nutrient retention [133]. These results support the need for further field studies of biochar to accurately assess its effects on soil quality and yield. Crop loss and other negative effects of biochar have also been demonstrated in some field trials over time. An increase in rapeseed yield was observed with more than 10 t ha^−1^ of biochar applied in the first year. A lower availability and reduction of organic matter in the soil can lead to less water [134].

Continued leaching of biochar-associated nutrients and a decrease in alkalinity are responsible for discoloration [23]. Another study found that the alkalinity associated with biochar was neutralized and the biochar converted to cations after three years of corn and grass field trials using Welsh sand clay [78]. Steiner et al. [73] also studied the effects of biochar over four Oxisol planting seasons (pH 4.5) in the Brazilian Amazon and found a cumulative yield increase of nearly 75% in rice and sorghum. Compared to a meta-analysis of more than 100 published studies, the average total yield increased by 13% due to biochar addition [135]. In addition, a meta-analysis of 103 publications found that pot experiments following the addition of biochar to the soil had the greatest effect on yield, which was greater in acidic soils (pH 5) than in field trials in neutral soils [135]. Likewise, a meta-analysis of 114 publications showed that the addition of biochar to soil significantly increased crop yields (about 20%) [19]. Overall, the meta-analysis studies suggest that biochar is a viable technology for yield enhancement in acidic and nutrient-poor soils.

### 4.7. Biochar and Regenerative Agriculture

Biochar, a carbon-rich material produced through the pyrolysis of organic matter, plays a pivotal role in advancing regenerative agriculture. Its multifaceted contributions enhance soil health, foster sustainable practices, and mitigate environmental challenges. Biochar serves as a stable carbon source, improving soil structure and nutrient retention. Its porous nature increases water-holding capacity, ensuring optimal moisture levels for plant growth. Moreover, biochar acts as a reservoir for essential nutrients, preventing leaching and promoting their availability to plants [136]. A crucial aspect of regenerative agriculture involves carbon sequestration to mitigate climate change. Biochar, being a carbon sink, aids in long-term carbon storage in soils by reducing atmospheric CO_2_ levels. This aligns with sustainable farming practices aimed at enhancing resilience to climate variability [137].

Biochar influences soil microbial communities, promoting beneficial microorganisms that contribute to nutrient cycling and plant health. It fosters a symbiotic relationship with mycorrhizal fungi, enhancing nutrient uptake by plant roots and improving overall soil biodiversity [136]. The adsorptive properties of biochar make it effective for the remediation of soil contaminants. It can immobilize heavy metals and pollutants, reducing their bioavailability and mitigating environmental risks. This capability contributes to sustainable land management and ecological restoration [138]. Biochar applications in agriculture can positively impact water quality. By preventing nutrient runoff and leaching, biochar helps maintain water purity, reducing the risk of nutrient pollution in water bodies. This aligns with regenerative agriculture practices that prioritize watershed health [139].

**Table 1 plants-13-00166-t001:** Impact of biochar on the growth, productivity, and yield of agronomic and horticultural crops.

Crop	Biochar Source	Application Rate	Effects	References
Cereal crops				
Rice	Wood biomass Forest wood biochar Wood residues Rice husk biochar	10 & 20% 5.5 & 11 t/ha 0–16 t/ha 4.13 g m^−2^	Prominent boost in yield and yield related attributes as compared to control	[37,54,129,140]
Wheat	Coppiced woodlands Hard & Soft wood Rice straw	30 & 60 t/ha 0 & 5% *w*/*w* 2 t/ha	Significant increase in growth and yield	[130,141,142]
Maize	Acacia bark Peanut hull	10 L m^−2^ 0 to 22.4 t/ha	Biochar application significantly improved the growth and yield of maize crop	[75,143]
Legume crops				
Mung bean	Maize straw	0–100 t/ha	25 t/ha resulted maximum grain yield	[144]
Common bean	Eucalyptus	10–50 t/ha	Biochar application boosted the yield of common bean	[131]
Oilseed crops				
Soybean	Oak tree	10 t/ha	Boost in dry matter yield	[145]
Oilseed rape	Litchi branch	10 to 30 t/ha	Positive impact on yield	[146]
Vegetable and tuber crops				
Sweet potato	Tobacco biochar Wheat straw	5 t/ha 0 to 40 t/ha	Improved vine length, tuber weight and yield	[147,148]
Potato	Biochar and N, P & K	7.5 t/ha	Higher tuber yield	[149]
Onion	Grass biochar	0.5 kg/m^2^	Boost in onion yield	[150]
Carrot	Wood biochar	0 to 20 t/acre	Positive impact on carrot weight	[151]

Studies indicate that the incorporation of biochar into agricultural soils can enhance crop yields. Its impact on nutrient availability, water retention, and soil structure contributes to increased productivity. Additionally, biochar-amended soils have demonstrated improved resilience to environmental stresses such as drought [64]. In summary, the integration of biochar in regenerative agriculture offers a holistic approach to sustainable land management. Its diverse benefits encompass soil fertility improvement, carbon sequestration, microbial support, contaminant remediation, water quality enhancement, and increased crop resilience.

## 5. Conclusions and Future Perspective

This review provides a comprehensive overview of biochar production and its characteristics, emphasizing its profound impacts on soil health and its pivotal roles in enhancing crop production and yields. The impacts of biochar on soil health are significant, with its ability to sequester carbon, improve water retention, and enhance nutrient availability. The reviewed literature emphasizes the positive effects of biochar on soil microbial communities, fostering a conducive environment for plant growth. In crop production, biochar stands out as a promising amendment, positively influencing plant growth, nutrient uptake, and overall crop yields. The intricate relationships between biochar characteristics, soil health improvement, and crop responses underline the importance of tailored approaches to its application. Prospects for biochar research and its application are promising yet challenging. Further investigations into the optimization of biochar production techniques, considering various feedstocks and pyrolysis conditions, will contribute to enhancing its efficacy. Long-term field studies are needed to unravel the nuanced interactions between biochar and soil-plant systems under diverse environmental conditions. Additionally, exploring the potential synergies of biochar with other agricultural practices and precision farming technologies could open new avenues for sustainable agriculture. In conclusion, the integration of biochar into agricultural systems holds great promise for addressing contemporary challenges in soil management, crop productivity, and environmental sustainability. Continued research efforts and collaborative initiatives are essential to unlock the full potential of biochar and pave the way for its widespread adoption in global agriculture.

## Data Availability

Data are contained within the article.

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
