# Peer review of "Biochar Production and Characteristics, Its Impacts on Soil Health, Crop Production, and Yield Enhancement: A Review"

_plants, 2024, doi:10.3390/plants13020166_

Round 1

Reviewer 1 Report

Comments and Suggestions for Authors

The article on the “Biochar Production and Characteristics, Impacts on Soil Health, and Roles in Crop Production and Yield Enhancement: A Review” is very interesting and informative with respect to biochar production and its application in agricultural crop productivity. The manuscript contains novel information which can further strengthen the existing knowledge of the field. Please consider addressing following concerns and incorporate suggestions before any consideration to publish this work.

Concerns

Title may be modified like “Biochar Production and Characteristics, its Impacts on Soil Health, Crop Production, and Yield Enhancement: A Review”.

Abstract

Line 12-13: Rewrite it, not clear.

In the abstract, add some mechanisms which are influenced by biochar.

Introduction

The introduction is well organized and may be improved with citations od recent studies.

Line 37: CO2, N2O, CH4. Proper formula.

Line 76: Typo mistake in title.

Biochar Production

More details may be included in the biochar production systems/processes.

Line 94-99: Typo mistakes are observed.

Biochar Impacts

4.6. Crop growth, development, and yield.

To specify the impact of biochar on crops, a table may be included with biochar impact on yield of various agronomic and horticultural crops.

Use journal’s guidelines for the format of references within text and bibliography.

Comments on the Quality of English Language

Moderate English Language Editing is required.

Author Response

Response Sheet

Subject:          Response to Comments

Title:   Biochar Production and Characteristics, its Impacts on Soil Health, Crop Production and Yield Enhancement: A Review.

We are very thankful to you for sparing time to review our manuscript and providing valuable comments and suggestions for the improvement of the manuscript.

I, Shahbaz Khan, corresponding author of the manuscript, am enclosing herewith a revised manuscript entitled “Biochar Production and Characteristics, its Impacts on Soil Health, Crop Production and Yield Enhancement: A Review” for publication in “Plants” after possible improvements. All the comments and suggestions are addressed accordingly and incorporated in the revised manuscript. Details of individual comments are given below.

Comment: Title may be modified like “Biochar Production and Characteristics, its Impacts on Soil Health, Crop Production, and Yield Enhancement: A Review”.

Response: Title is modified as suggested. The comments and suggestions are incorporated and highlighted in the revised manuscript.

Comment: Line 12-13: Rewrite it, not clear.

Response: The comments and suggestions are incorporated and highlighted in the revised manuscript.

Comment: In the abstract, add some mechanisms which are influenced by biochar.

Response: The comments and suggestions are incorporated and highlighted in the revised manuscript.

Comment: The introduction is well organized and may be improved with citations of recent studies.

Response: Thanks for complements. The comments and suggestions are incorporated and highlighted in the revised manuscript.

Comment: Line 37: CO2, N2O, CH4. Proper formula.

Response: The comments and suggestions are incorporated and highlighted in the revised manuscript.

Comment: Line 76: Typo mistake in title.

Response: The comments and suggestions are incorporated and highlighted in the revised manuscript.

Comment: More details may be included in the biochar production systems/processes.

Response: The comments and suggestions are incorporated and highlighted in the revised manuscript.

Comment: Line 94-99: Typo mistakes are observed.

Response: The comments and suggestions are incorporated and highlighted in the revised manuscript.

Comment: To specify the impact of biochar on crops, a table may be included with biochar impact on yield of various agronomic and horticultural crops.

Response: A table is added. The comments and suggestions are incorporated and highlighted in the revised manuscript.

Comment: Use journal’s guidelines for the format of references within text and bibliography.

Response: The comments and suggestions are incorporated and highlighted in the revised manuscript.

We say thanks again for your valuable comments and suggestions.

Reviewer 2 Report

Comments and Suggestions for Authors

The manuscript deals with an updated review on biochar as valuable organic source to improve the SOM management on agroecosystem. The Review provides suitable information on biochar production, and characteristics with a detailed and comprehensive discussion on the main effects of biochar on soil fertility and crop productivity

I highlighted minor issues that need to be faced before publication. Please refer to the comments in the  enclosed pdf document

Author Response

Response Sheet

Subject:          Response to Comments

Title:   Biochar Production and Characteristics, its Impacts on Soil Health, Crop Production and Yield Enhancement: A Review.

We are very thankful to you for sparing time to review our manuscript and providing valuable comments and suggestions for the improvement of the manuscript.

I, Shahbaz Khan, corresponding author of the manuscript, am enclosing herewith a revised manuscript entitled “Biochar Production and Characteristics, its Impacts on Soil Health, Crop Production and Yield Enhancement: A Review” for publication in “Plants” after possible improvements. All the comments and suggestions are addressed accordingly and incorporated in the revised manuscript. Details of individual comments are given below.

Comment: I would add Biochar as keyword.

Response: The comments and suggestions are incorporated and highlighted in the revised manuscript.

Comment: This statement about biochar seems generic; please try to give a more defined indication.

Response: Comprehensive definition is added. The comments and suggestions are incorporated and highlighted in the revised manuscript.

Comment: This looks a repetition of the previous statement on biochar properties.

Response: Repetition is removed. The comments and suggestions are incorporated and highlighted in the revised manuscript.

Comment: I do not understand this sentence about the sustainable use of biochar. What can vary between few and thousand of years?

Response: The comments and suggestions are incorporated and highlighted in the revised manuscript.

Comment: Why "On the contrary"?, the sentence is in line with the previous one on the beneficial or no negative effect of biochar application.

Response: The comments and suggestions are incorporated and highlighted in the revised manuscript.

Comment: The repetition of the terms "products, produced" is not formally incorrect, but it makes the reading not fluent- Perhaps you can find alternative terminology.

Response: The comments and suggestions are incorporated and highlighted in the revised manuscript.

Comment: The pollutants are not only heavy metals; moreover, the biochar is being characterized by a range methodologies other than elemental analysis and survey of functional groups (as detailed in the subsequent section). I would suggest rewriting the all paragraph.

Response: The comments and suggestions are incorporated and highlighted in the revised manuscript.

Comment: Line 155 -156 The meaning of the statement is not clear, Please rewrite.

Response: The comments and suggestions are incorporated and highlighted in the revised manuscript.

Comment: This paragraph is almost identical to the previous one; please merge them in a unique meaningful section.

Response: The comments and suggestions are incorporated and highlighted in the revised manuscript.

Comment: Also this paragraph is related to physical soil properites as the previous one (aggregates, porosity etc). I think it would be useful to have a unique well structured paragraph on the influence of biochar of soil physical properties.

Response: The comments and suggestions are incorporated and highlighted in the revised manuscript.

Comment: In the previous paragraps it is stated that the biochar increase the pH and the availability of nutrients (mainly base cations). By which mechanisms the biochar can reduce the sodicity and salinity? Please explain.

Response: The comments and suggestions are incorporated and highlighted in the revised manuscript.

Comment: I fully agree with the authors about the effect of biochar on CEC from the protonated functional groups. What is your opinion about the diffuse negative charges carried by the condensed polyaromatic components of biochar?

Response: The reviewer's comment aligns with the recognized impact of biochar on CEC primarily through protonated functional groups. Regarding the diffuse negative charges associated with condensed polyaromatic components in biochar, it's a pertinent observation. These aromatic structures contribute to the anion exchange capacity and can attract and hold onto negatively charged ions, further enhancing the overall ion exchange capacity of biochar. The combination of protonated functional groups and negative charges from aromatic components collectively influences the soil's nutrient retention and exchange dynamics, reinforcing the comprehensive role of biochar in soil improvement.

Comment: Improved P retention or P availability? Please check.

Response: The comments and suggestions are incorporated and highlighted in the revised manuscript.

We say thanks again for your valuable comments and suggestions.

Reviewer 3 Report

Comments and Suggestions for Authors

I appreciate the opportunity to review this paper as an effort to update my own knowledge on  impacts of biochar. This review was both useful and frustrating, but can be  a solid review article if issues below can be addressed. 

There are two modifications that would greatly enhance the quality of this effort.  First, the  list of analysis techniques to identify the composition of the biochar  (most of section 3) are currently NOT adequately connected to the rest of the manuscript.  We need to know why we want to do these analyses.  Unless the authors can do a road map from the specific analysis to the quality/functional behavior of the biochar, I would greatly reduce the section to perhaps two paragraphs total. Many soil scientists do not have extensive biochemical training, so including details of analysis techniques need to be defined in terms of simple soil response and biochar impact metrics. 

Second, if this is a review and does not present new research, the authors can write the paper in a consistent form, present tense, not past tense.  And, in conjunction with this I urge the authors to review existing  research summaries (some citations given below) to update their effort. I note very few papers written after 2020 that were cited here. A current review needs to incorporate and expand on the findings of previous reviews as well as individual studies.  In particular, that include material relevant to the environments being discussed need to be identified.  If this paper is to be cited, it needs to include at least an overview of recent additional reviews to justify the time it take to read it.  Also, we assume this article is not an ‘introductory text’ for non-scientists and can assume some basic knowledge of soils.  I’m your target audience: someone with some knowledge of agriculture and soils but needs to read an update of the status of biochar in developing sustainable agriculture.  Where is it particularly beneficial? Where does it show uncertainty? Portions of the text do this.

Given how much current emphasis is on regenerative agriculture and the role that biochar might play in this activity, it’s imperative that this review be up-to date.  For example, you state:  line 47: “it is still questionable whether the biochar applied to the soil is sustainable. According to studies, this can vary between a few years and thousands of years (Lehmann, 2007).”  Is there a more current reference that updates this contention? (and, you may want to include a definition of 'sustainable' here.  'Functional activity of biochar' is probably what the focus should be?)

Specific issues, suggested edits, comments:

Figure 1 is OK, but the right side of the figure can be reduced to two sentences that explain the benefit of biochar.  Why do we need to see the diagram of how biochar is made? This is not a college introductory text...

Line 16:  “The majority of biochar help the adsorption process with their highly porous structures and different functional groups.”  Clearer if you state: "The major benefit of biochar is...".

Line 22.  Before this sentence, I would add a sentence  like “Based upon our review, the greatest benefits to biochar appear to be….”., and if needed, clarify what environments this is limited to.  I think your last abstract sentence does not need to be said, and, if it was said, I would argue that we need ‘findings that inform agriculture sustainability efforts’, not more data!

Line 85:  you state  “It also has to be optimal.”  What is being optimized?

Lines 89-90.  Is this necessary?

Line 95.  Is the synthesis gas H or H2?

Line 134-5   What are the factors that control the pH of biochar?

Line 155  rewrite: “… a special feature resulting from the relationship between crystalline and amorphous phases.” 

Line 157  ‘Stability’ here refers to ‘longevity’ or period of functional activity?  Something can be stable when it is inert and no-reactive, yes?

 Line 201.  What material are we talking about here?  Is this now a research paper rather than review?

Line 182.  Shouldn’t the FTIR information be put together with that on lines starting 215?

Your definition of soil health is probably outdated.  I noted this one from Google:” Soil health is defined as the continued capacity of soil to function as a vital living ecosystem that sustains plants, animals, and humans. Healthy soil gives us clean air and water, bountiful crops and forests, productive grazing lands, diverse wildlife, and beautiful landscapes.”   Your definition ignores sustainability/resilience, etc. Your paper goes on to explain how biochar benefits those pieces of soil health identified in the updated definition.

Line 289:  you state “The use of biochar is essential…”  which isn’t true but “The use of biochar has the potential to greatly enhance soil health.” (and I've given up with trying to not get scientists to use the term 'soil health'.)

Line 408.  Please write out ‘Cation Exchange Capacity’ the first time it is identified, followed by ‘CEC’.

Line 422.  What are ‘exchangeable CIC cations’ ?

Line 427  “Especially during the tropical monsoon season, it leads to…”  what is ‘it’?

Line 505  “benthic fauna” = aquatic???  Soil fauna?

Line 546:  rewrite

Line 556-7:  Biochar has both positive and negative effects WITH RESPECT TO….  This paragraph needs to be edited to remove redundancy, and needs reorganization.  Please state the negatives and positives in a coherent manner.

Line 586   “…77%” of microbial biomass…

Line 653:  what is ‘it’?

Line 654.  2013 citation:  In today’s world, that’s not a “recent meta-analysis”

Concluding paragraph:  This is a disappointment.  We need to know the most updated facts about biochar that can be reported with high certainty.  We also need a sentence to identify those things that you say need to be measured with the new techniques.

Not all references are alphabetical  (c.f. Line 891)

References that should be considered in relating their findings to this current synthesis effort:

Joseph, S., Cowie, A.L., Van Zwieten, L., Bolan, N., Budai, A., Buss, W., Cayuela, M.L., Graber, E.R., Ippolito, J.A., Kuzyakov, Y. and Luo, Y., 2021. How biochar works, and when it doesn't: A review of mechanisms controlling soil and plant responses to biochar. Gcb Bioenergy, 13(11), pp.1731-1764.

Brtnicky, M., Datta, R., Holatko, J., Bielska, L., Gusiatin, Z.M., Kucerik, J., Hammerschmiedt, T., Danish, S., Radziemska, M., Mravcova, L. and Fahad, S., 2021. A critical review of the possible adverse effects of biochar in the soil environment. Science of the Total Environment, 796, p.148756.

Kamali, M., Sweygers, N., Al-Salem, S., Appels, L., Aminabhavi, T.M. and Dewil, R., 2022. Biochar for soil applications-sustainability aspects, challenges and future prospects. Chemical Engineering Journal, 428, p.131189.

Liu, Z., Zhu, M., Wang, J., Liu, X., Guo, W., Zheng, J., Bian, R., Wang, G., Zhang, X., Cheng, K. and Liu, X., 2019. The responses of soil organic carbon mineralization and microbial communities to fresh and aged biochar soil amendments. GCB bioenergy, 11(12), pp.1408-1420.

Zhang, Y., Wang, J. and Feng, Y., 2021. The effects of biochar addition on soil physicochemical properties: A review. Catena, 202, p.105284.

Comments on the Quality of English Language

included in comments above

Author Response

Response Sheet

Subject:          Response to Comments

Title:   Biochar Production and Characteristics, its Impacts on Soil Health, Crop Production and Yield Enhancement: A Review.

We are very thankful to you for sparing time to review our manuscript and providing valuable comments and suggestions for the improvement of the manuscript.

I, Shahbaz Khan, corresponding author of the manuscript, am enclosing herewith a revised manuscript entitled “Biochar Production and Characteristics, its Impacts on Soil Health, Crop Production and Yield Enhancement: A Review” for publication in “Plants” after possible improvements. All the comments and suggestions are addressed accordingly and incorporated in the revised manuscript. Details of individual comments are given below.

Comment: I appreciate the opportunity to review this paper as an effort to update my own knowledge on impacts of biochar. This review was both useful and frustrating, but can be a solid review article if issues below can be addressed.

Response: Thank you for investing your time and energy to review the article critically. We have tried our best to incorporate the concerns accordingly.

Comment: There are two modifications that would greatly enhance the quality of this effort.  First, the  list of analysis techniques to identify the composition of the biochar  (most of section 3) are currently NOT adequately connected to the rest of the manuscript.  We need to know why we want to do these analyses.  Unless the authors can do a road map from the specific analysis to the quality/functional behavior of the biochar, I would greatly reduce the section to perhaps two paragraphs total. Many soil scientists do not have extensive biochemical training, so including details of analysis techniques need to be defined in terms of simple soil response and biochar impact metrics.

Response: The comments and suggestions are incorporated and highlighted in the revised manuscript.

Comment: Second, if this is a review and does not present new research, the authors can write the paper in a consistent form, present tense, not past tense.  And, in conjunction with this I urge the authors to review existing  research summaries (some citations given below) to update their effort. I note very few papers written after 2020 that were cited here. A current review needs to incorporate and expand on the findings of previous reviews as well as individual studies.  In particular, that include material relevant to the environments being discussed need to be identified.  If this paper is to be cited, it needs to include at least an overview of recent additional reviews to justify the time it take to read it.  Also, we assume this article is not an ‘introductory text’ for non-scientists and can assume some basic knowledge of soils.  I’m your target audience: someone with some knowledge of agriculture and soils but needs to read an update of the status of biochar in developing sustainable agriculture.  Where is it particularly beneficial? Where does it show uncertainty? Portions of the text do this.

Response: Suggested citations are included. Suggested articles are cited. The comments and suggestions are incorporated and highlighted in the revised manuscript.

Comment: Given how much current emphasis is on regenerative agriculture and the role that biochar might play in this activity, it’s imperative that this review be up-to date.  For example, you state:  line 47: “it is still questionable whether the biochar applied to the soil is sustainable. According to studies, this can vary between a few years and thousands of years (Lehmann, 2007).”  Is there a more current reference that updates this contention? (and, you may want to include a definition of 'sustainable' here.  'Functional activity of biochar' is probably what the focus should be?).

Response: Details regarding regenerative agriculture with respect to biochar are added (4.7. Biochar and Regenerative Agriculture). The comments and suggestions are incorporated and highlighted in the revised manuscript.

Comment: Figure 1 is OK, but the right side of the figure can be reduced to two sentences that explain the benefit of biochar.  Why do we need to see the diagram of how biochar is made? This is not a college introductory text.

Response: Figure is removed.

Comment: Line 16: “The majority of biochar help the adsorption process with their highly porous structures and different functional groups.”  Clearer if you state: "The major benefit of biochar is...".

Response: The comments and suggestions are incorporated and highlighted in the revised manuscript.

Comment: Line 22.  Before this sentence, I would add a sentence like “Based upon our review, the greatest benefits to biochar appear to be….”., and if needed, clarify what environments this is limited to.  I think your last abstract sentence does not need to be said, and, if it was said, I would argue that we need ‘findings that inform agriculture sustainability efforts’, not more data!.

Response: The comments and suggestions are incorporated and highlighted in the revised manuscript.

Comment: Line 85:  you state “It also has to be optimal.”  What is being optimized?

Response: The comments and suggestions are incorporated and highlighted in the revised manuscript.

Comment: Lines 89-90.  Is this necessary?

Response: The comments and suggestions are incorporated and highlighted in the revised manuscript.

Comment: Line 95.  Is the synthesis gas H or H2?

Response: The comments and suggestions are incorporated and highlighted in the revised manuscript.

Comment: Line 134-5   What are the factors that control the pH of biochar?

Response: The comments and suggestions are incorporated and highlighted in the revised manuscript.

Comment: Line 155 rewrite: “… a special feature resulting from the relationship between crystalline and amorphous phases”.

Response: The comments and suggestions are incorporated and highlighted in the revised manuscript.

Comment: Line 157 ‘Stability’ here refers to ‘longevity’ or period of functional activity?  Something can be stable when it is inert and no-reactive, yes?

Response: The comments and suggestions are incorporated and highlighted in the revised manuscript.

Comment: Line 201.  What material are we talking about here?  Is this now a research paper rather than review?

Response: Thanks for complements. The comments and suggestions are incorporated and highlighted in the revised manuscript.

Comment: Line 182.  Shouldn’t the FTIR information be put together with that on lines starting 215?

Response: The comments and suggestions are incorporated and highlighted in the revised manuscript.

Comment: Your definition of soil health is probably outdated.  I noted this one from Google:” Soil health is defined as the continued capacity of soil to function as a vital living ecosystem that sustains plants, animals, and humans. Healthy soil gives us clean air and water, bountiful crops and forests, productive grazing lands, diverse wildlife, and beautiful landscapes.”   Your definition ignores sustainability/resilience, etc. Your paper goes on to explain how biochar benefits those pieces of soil health identified in the updated definition.

Response: The comments and suggestions are incorporated and highlighted in the revised manuscript.

Comment: Line 289:  you state “The use of biochar is essential…”  which isn’t true but “The use of biochar has the potential to greatly enhance soil health.” (and I've given up with trying to not get scientists to use the term 'soil health'.).

Response: The comments and suggestions are incorporated and highlighted in the revised manuscript.

Comment: Line 408.  Please write out ‘Cation Exchange Capacity’ the first time it is identified, followed by ‘CEC’.

Response: The comments and suggestions are incorporated and highlighted in the revised manuscript.

Comment: Line 422.  What are ‘exchangeable CIC cations’?

Response: The comments and suggestions are incorporated and highlighted in the revised manuscript.

Comment: Line 427 “Especially during the tropical monsoon season, it leads to…”  what is ‘it’?

Response: The comments and suggestions are incorporated and highlighted in the revised manuscript.

Comment: Line 505 “benthic fauna” = aquatic???  Soil fauna?

Response: Title is modified as suggested. The comments and suggestions are incorporated and highlighted in the revised manuscript.

Comment: Line 546:  rewrite.

Response: The comments and suggestions are incorporated and highlighted in the revised manuscript.

Comment: Line 556-7:  Biochar has both positive and negative effects WITH RESPECT TO….  This paragraph needs to be edited to remove redundancy and needs reorganization.  Please state the negatives and positives in a coherent manner.

Response: The comments and suggestions are incorporated and highlighted in the revised manuscript.

Comment: Line 586   “…77%” of microbial biomass.

Response: Thanks for complements. The comments and suggestions are incorporated and highlighted in the revised manuscript.

Comment: Line 653:  what is ‘it’?

Response: The comments and suggestions are incorporated and highlighted in the revised manuscript.

Comment: Line 654.  2013 citation:  In today’s world, that’s not a “recent meta-analysis”.

Response: The comments and suggestions are incorporated and highlighted in the revised manuscript.

Comment: Concluding paragraph:  This is a disappointment.  We need to know the most updated facts about biochar that can be reported with high certainty.  We also need a sentence to identify those things that you say need to be measured with the new techniques.

Response: The comments and suggestions are incorporated and highlighted in the revised manuscript.

Comment: Not all references are alphabetical (c.f. Line 891).

Response: References are formatted according to journal styles and author guidelines. The comments and suggestions are incorporated and highlighted in the revised manuscript.

Comment: References that should be considered in relating their findings to this current synthesis effort.

Response: Suggested references are cited accordingly. The comments and suggestions are incorporated and highlighted in the revised manuscript.

We say thanks again for your valuable comments and suggestions.

Reviewer 4 Report

Comments and Suggestions for Authors

Dear Authors,

Your work is a reasonable concept of collecting information on the application of biocarbon in agriculture. The content as well as the layout of the work is correctly preserved. The literature review has been presented quite extensively, although one might still be tempted to expand the existing literature with research on biocarbon conducted by Prof. Agnieszka Latawiec.

There were also some errors in the paper that should be corrected, here they are:

- line 90-91: "After all, you can lose weight by 90 burning leftover carbon" maybe better is write "After all, it can...carbon"

- line 146-147: "The disadvantage of this method is that underestimation of 146 ash will lead to overestimation of cabbage" - this sentence is unintelligible, please correct it

- line 195: "market area" is this form correct?

- line 292: The entire section "4.1 Soil Health" in the section "4.2 Pchysical properties..." is repeated. (line 316)

- line 372: two consecutive sentences begin the same way "In addition..." - this should be corrected

- line 555: the word "bacteria" is repeated.

Besides, please standardize the writing of subscripts in the publication when writing out chemical formulas like CO2, CH4, N20, H2. Please also standardize the order of writing the year of publication in the bibliography.

After correcting the above-mentioned remarks, the work can be accepted for publication.

Author Response

Response Sheet

Subject:          Response to Comments

Title:   Biochar Production and Characteristics, its Impacts on Soil Health, Crop Production and Yield Enhancement: A Review.

We are very thankful to you for sparing time to review our manuscript and providing valuable comments and suggestions for the improvement of the manuscript.

I, Shahbaz Khan, corresponding author of the manuscript, am enclosing herewith a revised manuscript entitled “Biochar Production and Characteristics, its Impacts on Soil Health, Crop Production and Yield Enhancement: A Review” for publication in “Plants” after possible improvements. All the comments and suggestions are addressed accordingly and incorporated in the revised manuscript. Details of individual comments are given below.

Comment: Your work is a reasonable concept of collecting information on the application of biocarbon in agriculture. The content as well as the layout of the work is correctly preserved. The literature review has been presented quite extensively, although one might still be tempted to expand the existing literature with research on biocarbon conducted by Prof. Agnieszka Latawiec.

Response: Thank you so much for your complements. The work by Prof. Agnieszka Latawiec is also cited in the revised manuscript.

Comment: line 90-91: "After all, you can lose weight by burning leftover carbon" maybe better is write "After all, it can...carbon".

Response: The comments and suggestions are incorporated and highlighted in the revised manuscript.

Comment: line 146-147: "The disadvantage of this method is that underestimation of ash will lead to overestimation of cabbage" - this sentence is unintelligible, please correct it.

Response: The comments and suggestions are incorporated and highlighted in the revised manuscript.

Comment: line 195: "market area" is this form correct?

Response: The comments and suggestions are incorporated and highlighted in the revised manuscript.

Comment: line 292: The entire section "4.1 Soil Health" in the section "4.2 Pchysical properties..." is repeated. (line 316).

Response: The comments and suggestions are incorporated and highlighted in the revised manuscript.

Comment: line 372: two consecutive sentences begin the same way "In addition..." - this should be corrected.

Response: The comments and suggestions are incorporated and highlighted in the revised manuscript.

Comment: line 555: the word "bacteria" is repeated.

Response: The comments and suggestions are incorporated and highlighted in the revised manuscript.

Comment: Besides, please standardize the writing of subscripts in the publication when writing out chemical formulas like CO2, CH4, N20, H2. Please also standardize the order of writing the year of publication in the bibliography.

Response: The comments and suggestions are incorporated and highlighted in the revised manuscript.

Comment: After correcting the above-mentioned remarks, the work can be accepted for publication.

Response: All the comments have been addressed accordingly.

We say thanks again for your valuable comments and suggestions.

Round 2

Reviewer 3 Report

Comments and Suggestions for Authors

I apologize if I seem to get a bit terse regarding this review.  This paper attempts an overview on a topic – a title that attempts to link biochar to plant productivity and soil health – that potentially requires 1000s of pages to accomplish.  Something more extracted – more condensed – is a difficult task.  The paper attempts too much, and in doing so frustrates readers who want a succinct summary and status report.

As written, this paper is largely not useful for readers.  I, like most, am a researcher working on an applied subject such as crop productivity and soil health, therefore I’m looking for information that’s relevant to what I do.  Instead, the article dives into carbon biochemistry while not telling us why we should care, and then it gives us an introductory lecture on soils that made me go look at the title again.  (this is about biochar and soils…not soils… I understand soils and plant productivity….I want to know a state-of-the-art summary of how biochar can impact this. ) The manuscript repeats statements that are not needed even the first time they were presented.  This revision attempted a band-aid response to the original review, which really requested a major rewrite and reorganization. 

I gave up reading this somewhere around line 400 where redundant statements frustrated me beyond belief. 

I had to wonder when reading this:  Is this an attempt to publish an introductory chapter of a dissertation?  Is this paper written by artificial intelligence and is a test to see if reviewers can detect this?  (the almost random restatements make one think this.)

One issue I did conclude this that…for journal research reviews…the topic must be simplified.  Since soils and climate drive biochar impacts, perhaps simply focusing on a subset of the planet is essential to be successful in this effort.

Paragraph, lines 122 to 136.  Your audience cannot appreciate any of this because they cannot link it to impacts on plant productivity.

Line 81.  Sentence not needed…and distracting in that you’ve told us this, already.

Lin3 126:  “hdrocarbons’ is an incredibly broad term…  Probably need to clarify.

Line 150.  Decomposition or oxidation?

Line 172.  Not clear

Line  207  the units I’m familiar with for bulk density are grams/cubic cm.

Author Response

Response Sheet

Subject:          Response to Comments

Title:   Biochar Production and Characteristics, its Impacts on Soil Health, Crop Production and Yield Enhancement: A Review.

We are very thankful to you for sparing time to review our manuscript and providing valuable comments and suggestions for the improvement of the manuscript.

I, Shahbaz Khan, corresponding author of the manuscript, am enclosing herewith a revised manuscript entitled “Biochar Production and Characteristics, its Impacts on Soil Health, Crop Production and Yield Enhancement: A Review” for publication in “Plants” after possible improvements. All the comments and suggestions are addressed accordingly and incorporated in the revised manuscript. Details of individual comments are given below.

Comment: I apologize if I seem to get a bit terse regarding this review.  This paper attempts an overview on a topic – a title that attempts to link biochar to plant productivity and soil health – that potentially requires 1000s of pages to accomplish.  Something more extracted – more condensed – is a difficult task.  The paper attempts too much, and in doing so frustrates readers who want a succinct summary and status report.

As written, this paper is largely not useful for readers.  I, like most, am a researcher working on an applied subject such as crop productivity and soil health, therefore I’m looking for information that’s relevant to what I do.  Instead, the article dives into carbon biochemistry while not telling us why we should care, and then it gives us an introductory lecture on soils that made me go look at the title again.  (this is about biochar and soils…not soils… I understand soils and plant productivity….I want to know a state-of-the-art summary of how biochar can impact this. ) The manuscript repeats statements that are not needed even the first time they were presented.  This revision attempted a band-aid response to the original review, which really requested a major rewrite and reorganization. 

I gave up reading this somewhere around line 400 where redundant statements frustrated me beyond belief. 

I had to wonder when reading this:  Is this an attempt to publish an introductory chapter of a dissertation?  Is this paper written by artificial intelligence and is a test to see if reviewers can detect this?  (the almost random restatements make one think this.)

One issue I did conclude this that…for journal research reviews…the topic must be simplified.  Since soils and climate drive biochar impacts, perhaps simply focusing on a subset of the planet is essential to be successful in this effort.

Response: We appreciate your concerns and views. In this review, we are not limited to crop productivity and soil health. This review covers the production process and properties of biochar, and provides comprehensive details about the biochar. That is why we tried to cover most of the aspects of biochar.

Comment: Paragraph, lines 122 to 136.  Your audience cannot appreciate any of this because they cannot link it to impacts on plant productivity.

Response: You may be right as your concern is crop productivity. We included this part to provide information on biochar production. Before the application of biochar, it production is mandatory. It is one of the processes to produce the biochar.

Comment: Line 81.  Sentence not needed…and distracting in that you’ve told us this, already.

Response: Deleted. The comments and suggestions are incorporated and highlighted in the revised manuscript.

Comment: Lin3 126: “hdrocarbons’ is an incredibly broad term…  Probably need to clarify.

Response: The lignin components that are not dissolved in liquid phase are transferred into hydrochar like pyrolysis process. The comments and suggestions are incorporated and highlighted in the revised manuscript.

Comment: Line 150.  Decomposition or oxidation?

Response: Oxidation. The comments and suggestions are incorporated and highlighted in the revised manuscript.

Comment: Line 172.  Not clear.

Response: Deleted. The comments and suggestions are incorporated and highlighted in the revised manuscript.

Comment: Line 207 the units I’m familiar with for bulk density are grams/cubic cm.

Response: Unit is updated. The comments and suggestions are incorporated and highlighted in the revised manuscript.

We say thanks again for your valuable comments and suggestions.